# Water Sink Model for Robot Motion Planning

**DOI:** 10.3390/s19061269

**Published:** 2019-03-13

**Authors:** Gi-Yoon Jeon, Jin-Woo Jung

**Affiliations:** 1Department of Computer Science and Engineering, Dongguk University, 30, Pildong-Ro 1-Gil, Jung-Gu, Seoul 04620, Korea; gyjeon@add.re.kr; 2Agency for Defense Development, Songpa P.O. Box 132, Seoul 05771, Korea

**Keywords:** robot motion planning, water sink model, artificial potential field, local minima

## Abstract

There are various motion planning techniques for robots or agents, such as bug algorithm, visibility graph, Voronoi diagram, cell decomposition, potential field, and other probabilistic algorithms. Each technique has its own advantages and drawbacks, depending on the number and shape of obstacles and performance criteria. Especially, a potential field has vector values for movement guidance to the goal, and the method can be used to make an instantaneous and smooth robot movement path without an additional controller. However, there may be some positions with zero force value, called local minima, where the robot or agent stops and cannot move any further. There are some solutions for local minima, such as random walk or backtracking, but these are not yet good enough to solve the local minima problem. In this paper, we propose a novel movement guidance method that is based on the water sink model to overcome the previous local minima problem of potential field methods. The concept of the water sink model is to mimic the water flow, where there is a sink or bathtub with a plughole and floating piece on the water. The plughole represents the goal position and the floating piece represents robot. In this model, when the plug is removed, water starts to drain out via the plughole and the robot can always reach the goal by the water flow. The water sink model simulator is implemented and a comparison of experimental results is done between the water sink model and potential field.

## 1. Introduction

There are a number of motion planning techniques for robots or agents. Bug algorithms are very intuitive and simple, but inefficient in many cases [1], while visibility graphs can guarantee the shortest path on a two-dimensional (2D) map, but make low clearance from obstacles and are only good for polygonal obstacle environments [2]. Moreover, Voronoi diagrams can guarantee high clearance from obstacles, but they also have high computational complexity for path creation [3], and cell decomposition is simple but also carrying high computational complexity depending on the number of created cells [4]. The PRM (Probabilistic Roadmap Method) and RRT (Rapidly exploring Random Tree) are probabilistic algorithms. PRM makes a graph with node that is randomly generated and find the shortest path on the graph [5,6]. The RRT makes a tree that is randomly expanded from the starting point to the goal. The RRT provides a successful result in many cases. However, the RRT could not guarantee the optimality and completeness of path. The randomized kinodynamic planning considers velocity, acceleration, force, and torque of mobile robot with original RRT [7]. The RRT* is a mixture of PRM and RRT and it generates a random graph to find a path [8] and the informed RRT* is an improvement method for the optimality and completeness of RRT* [9]. In recent years, the heuristic-based algorithm, including neural network, fuzzy logic technique, and nature inspired algorithms has been proposed [10].

Most of these methods require an additional controller in the physical application to follow the created path. Besides, the potential field method can make an instantaneous and smooth robot movement path without an additional controller, but it has a local minima problem that is caused by positions with zero force value where the robot or agent stops and cannot move any further [11,12]. There are some techniques to overcome the local minima problem, such as random walk [13,14] or backtracking. In the case of the random walk technique, it is impossible to predict the path or the processing time because of randomness. The potential guided directional-RRT* [15] is a hybrid method of RRT* and artificial potential field, where artificial potential field directs the random samples toward the goal. This leads to an increase in the speed of RRT*. However, this is still one of the RRT methods. Therefore, this method does not guarantee completeness. In addition, there is an adaptive motion planning method for dynamic environment with artificial potential field using an RRT-based prior path between the starting position and the goal position [16]. However, this is still one of potential field methods. Therefore, this method is not completely free from the local minima problem and inefficiency because this method requires two kinds of path planning processes. To reduce the processing time of the original potential field method, there is a technique by replacing complex obstacles to bounding circles for high-speed vehicles [17]. Similarly, the processing time of potential field method depends on the number of sampling points or the resolution of configuration space. Therefore, a method for reducing the processing time of the potential field method using non-uniform cell decomposition has also been proposed [18]. Even though many variants of the potential field method have been proposed, the local minima problem was not completely resolved yet.

In this paper, we propose a new robot motion planning method, namely water sink model-based motion planning, to resolve the local minima problem using the potential field method by mimicking the flow of water draining out of a sink through a plughole. In the experimental results, the water sink model is compared with the potential field method and the results show a sufficiently practical path on the grid map.

## 2. Related Works

### 2.1. Potential Field

A robot in a potential field can be treated as a point, i.e., the robot is represented in the configuration space as a particle under the influence of an artificial potential field U, whose variations reflect the structure of the free space U(q) for given configuration q.

The potential function can be defined over the free space as the sum of an attraction potential $U_{att}\left(q\right)$ pulling the robot towards the goal configuration and a repulsion potential $U_{rep}\left(q\right)$ pushing the robot away from the obstacles [10], and it can be expressed as
(1)$$U\left(q\right)=U_{att}\left(q\right)+U_{rep}\left(q\right)$$
which is visually exhibited in Figure 1 [12]. The $U_{att}\left(q\right)$ is attractive forces (see Figure 1a), the $U_{rep}\left(q\right)$ is repulsive forces (see Figure 1b), and U(q) is the sum of attractive forces and repulsive forces (see Figure 1c). The attraction and repulsion potential functions can be formalized, as follows:(2)$$U{\left(q\right)}_{att}=\frac{1}{2}\xi ∥q-q_{goal}\;{∥}^{2}$$
(3)$$U_{rep}\left(q\right)=\{\begin{array}{cc} \frac{1}{2}\eta {\left(\frac{1}{\rho \left(q\right)}-\frac{1}{\rho_{0}}\right)}^{2}, & \rho \left(q\right)\leq \rho_{0}\\ 0, & \rho \left(q\right)\geq \rho_{0} \end{array}$$

In Equation (2), ξ is a scaling factor of attraction force and $q_{goal}$ is the position of goal. In Equation (3), η is a scaling factor of repulsive force between robot and obstacles. ρ(q) is the minimum value of distance from the obstacles and $\rho_{0}$ is the influent distance.

### 2.2. Analysis of Potential Field Method

The potential field method is a kind of combination model for robot path planning and control. The control input for the next robot movement is calculated while using the force field that is the negative gradient of the sum of the potential fields, thus a robot can move smoothly and instantly on the field. However, the sum of attraction and repulsion potentials may have zero vector values at some positions (local minima—see Figure 2), where the robot cannot move to any direction.

There have been some suggested improvements to resolve the local minima problem, with one method being random walk. If a robot is in a local minimum position, the robot tries to walk in a random direction to escape the local minimum [15]. However, this method has no guarantee of arrival at the goal on every occasion.

## 3. Water Sink Model-Based Motion Planning

### 3.1. The Main Idea behind the Water Sink Model

The concept of the water sink model is to consider the field as a sink or bathtub with a plughole, in which the sink is assumed to be filled with enough water at the initial stage, the robot is assumed to be a floating piece, and the goal is assumed to be the plughole. In this model, when the plug is removed, water starts to drain out via the plughole, and ideally the water supply cannot be exhausted. The main idea is that there are no local minima that are caused by the flow mechanism of the water and the robot is always able to reach the goal (see Figure 3).

### 3.2. Water Sink Model Generation

For the implementation of the water sink model, a grid is made of the field in four directions (up, down, left, right) out of cells, the number of plugholes is assumed to be one and it is always included, the volume of draining water via the plughole over time is assumed to be constant, and the number of cells that are involved in water flow expands as time passes. Water flow is assumed to occur from a cell with a higher water level to a cell with a lower water level (due to gravity) at each discrete time instance.

First, only one grid cell is involved in draining. During the next time instance, the first cell and its four connected ones are used together for water drainage, then the number of cells for water drainage continuously expands and the water volume per cell for drainage decreases as time passes. In Figure 4, the assumed draining water volume per one time instance is assumed as 100. Algorithm 1 reflects the WaterMapGeneration algorithm and Figure 5 is an execution example for the water sink model. The example consists of 5 × 5 cells. The cell number 7 is plughole and the cell number 5, 9, 10, 14, and 15 are obstacles. The color of the cell means the water level. The color of the cell becomes brighter when the water level is lower. Contrarily, the color of the cell becomes darker when the water level is higher. The propagation velocity of water drainage, $v_{prop}$ should be defined for the water sink model. In the case of Figure 4, the $v_{prop}$ is assumed as 1.

**Algorithm 1.** WaterMapGeneration1:**procedure** generateWaterMap(plughole, robot, wall, drainList, waterMap)2: **if** drainList is empty or robot is in drainList3:  **return** waterMap4: cellNumber ← the size of drainList.5: drain ← drainage water per 1 turn/cellNumber.6: index ← 0.7: **repeat**8:  remove water from waterMap[drainList[index]].9:  insertDrain(four neighborhood cells of waterMap[drainList[index]], wall, drainList).10: **until** the end of drainList11: **return** waterMap12:**procedure** insertDrain(drainCell, wall, drainList)13: **if** drainList is empty or drainCell is in drainList14:  **return** drainList15: end ← last cell index of drainList.16: drainList[end] ← drainCell.17: **return** drainList

The complexity of WaterMapGeneration algorithm is $O\left(n^{2}\right)$. Here, n is the number of cells in the grid. Water drainage occurs at every turn by propagating to the adjacent cells. The searching continues until the drainList includes the robot cell where the robot exists. In the worst case, the robot cell will be inserted to the drainList at the *n*th turn.

### 3.3. Water Sink Model-Based Path Generation

Water drainage extends to the adjacent four cells (up, down, left, and right), but the robot can move in any of the adjacent eight directions. If there are two or more cells with the same level of water, then the robot can move to any of these, as cells with the same level of water means an equal chance for the robot to move to any one of them. The following Algorithm 2 represents a pseudocode for path-finding in water sink model:

**Algorithm 2.** RobotPathGeneration1:**procedure** generateRobotPath(plughole, robot, waterMap, robotPath)2: **if** plughole position is same robot3:  **return** robotPath4: index ← 0.5: **repeat**6:  tempCell[0 …7] ← eight neighborhood cells of robot.7:  robot ← minimum water value cell of tempCell[0…7].8:  robotPath[index] ← robot.9:  index ← index + 1.10: **until** robot meets plughole11: **return** robotPath

Figure 6 shows two cases of path generation when the robot is positioned at the cell number 23 in Figure 5d. Figure 6a shows that the water sink model considers four-connectivity for path generation. The robot is able to select the cell number 18 or 22. Regardless of the selection, the lengths of the generated paths are equivalent. However, the four-connectivity model may generate a zigzag motion. To avoid the zigzag motion and enhance the efficiency, the eight-connectivity model is applied in the water sink model. In Figure 6b, the generated sequence of path is cell number 23, 17, 12, and 7.

The concept of the water sink model may be looked at similarly with Wavefront path planning methods. However, the Wavefront method [19,20], Modified Wavefront method [19], and Focused Wavefront method [19] use grid cells with a natural number and that number is only utilized for path finding. However, the water sink model uses the real number for the water level in the grid cell. The robot controller or actuator uses this real value to determine the speed as well as the direction of robot movement as the potential field method. In addition, in various kinds of Wavefront methods, all grid cells’ natural numbers are fixed, but in the water sink model, the water levels are changed by turn by turn and the amount of water that is drained at a time can be adjusted for various situations.

The complexity of the RobotPathGeneration algorithm is O(n). Here, the n is the number of cells in drainList. The last data of drainList is the index number of the robot cell and the first data of drainList is the index number of the plughole cell. This algorithm generates robotPath by traversing cell numbers from the last data to the first data of drainList. Since the water level could be searched with the cell number at a constant time, the traversing is also completed in O(n).

### 3.4. Mathematical Verification

In this chapter, we attempt to mathematically verify our method. We explain the simple water model without obstacles and then consider the case with them.

**Theorem** **1.**
*In the waterMap generated from WaterMapGeneration algorithm, the water level of a cell farther from the plughole is always higher than the one closer to the plughole if there are no obstacles.*


**Proof of Theorem** **1.**First, we calculate the population of cells that are involved in drainage for each turn (drainage is performed as in Figure 4) and seek for the recurrent equation.The population of cells that are related with first turn drainage is 1, the population of cells related with second turn drainage are 1 + 4 = 5, the population of cells related with third turn drainage are 1 + 4 + 8 = 13, and the population of cells related with fourth turn drainage are 1 + 4 + 8 + 12 = 25, and so on.The population of boundary cells for drainage is increased to 0, 4, 8, 12, … and the population of boundary cells for *n*th drainage are
(4)4n−4,n=1,2,3, …
and the population of cells for total water drainage until *n*th turn are
(5)$$\begin{array}{l} a_{n}=a_{n-1}+4n-4,\;a_{1}=1\\ =\overset{n}{\underset{k=1}{\sum }}\left(4k-4\right)+a_{1}=4\times \frac{n\left(n-1\right)}{2}+1=2n\left(n-1\right)+1\\ =2n^{2}-2n+1 \end{array}$$Let *d* be a distance to the cell from the plughole and the distance is measured by city-block or taxicab geometry. Let $V_{d}\left(t\right)$ be the water level in the cell at d distance from the plughole for t turn $\left(0\leq d\leq v_{prop}t\right)$. Let $V_{0}$ be the initial water level in each cell and C be the amount of drainage water per turn. When the cell with distance $d_{k}$ is concerned with the drainage for the first time, $\frac{C}{2d_{k}^{2}-2d_{k}+1}$ of water will fall out of the cell. At the next turn, the cells with distance $d_{k}$ and $d_{k+1}$ are concerned in the drainage and $\frac{C}{2d_{k}^{2}-2d_{k}+1}+\frac{C}{2d_{k+1}^{2}-2d_{k+1}+1}$ of water will fall out of the cells. Therefore, the remained water level of cell until t turn with distance $d_{1}$ and $d_{2}$ are expressed by $V_{d_{1}}\left(t\right),V_{d_{2}}\left(t\right)$, and $\left(0\leq d_{1}<d_{2}\leq v_{prop}t\right)$, as
(6)$$V_{d_{1}}\left(t\right)=V_{0}-\left(\frac{C}{a_{d_{1}}}+\frac{C}{a_{d_{1}+1}}+\cdots +\frac{C}{a_{d_{2}}}+\frac{C}{a_{d_{2}+1}}+\cdots +\frac{C}{a_{t}}\right),\;v_{prop}=1$$
(7)$$V_{d_{2}}\left(t\right)=V_{0}-\left(\frac{C}{a_{d_{2}}}+\frac{C}{a_{d_{2}+1}}+\cdots +\frac{C}{a_{t}}\right),\;v_{prop}=1$$Subsequently, the water level of $V_{d_{2}}\left(t\right)$ is always smaller than $V_{d_{1}}\left(t\right)$, and so
(8)$$V_{d_{1}}\left(t\right)<V_{d_{2}}\left(t\right)$$ □

**Theorem** **2.**
*There are no local minima on the robot path from RobotPathGeneration algorithm in water sink model.*


**Proof of Theorem** **2.**Assume that the robot is located at a local minimum position, then without the loss of generality, this position can be regarded as the new robot starting position. Before finding the robot path, the water sink model has already been created with a plughole while using the above WaterMapGeneration algorithm. Therefore, there is a path that connects the goal to the robot starting position.If there is a local minimum in the water sink model, then the robot cannot move in any direction from this position. Therefore, there is no difference in the water level between neighboring cells and there is no water drainage thereafter. When a grid cell has no water drainage, the neighboring grid cells also cannot have water drainage. Consequently, the number of grid cells with no water drainage is expanded until some of them meet with the plughole cell. As result, water cannot drain out of the plughole, although some water is able to drain through the plughole during every discrete time instance, which is contradictory. Therefore, the first assumption that a robot located at a local minimum position is false, and so there is no local minimum on the robot path. □

**Theorem** **3.**
*RobotPathGeneration algorithm always generate a path from the robot position to the goal if the robot or plughole is not surrounded by obstacles.*


**Proof of Theorem** **3.**If we consider backtracking from the goal position (plughole) to the start position (robot), a cell that is adjacent to the robot position always exists and the cell is closer to the goal by a single cell length. After the robot moves to an adjacent cell in the same manner, a closer cell to the goal exists by a single cell length once again. This process is continued until the robot reaches the goal. Therefore, if the goal or robot is not surrounded by obstacles, there exists a path from the robot to the plughole on every occasion. □

**Theorem** **4.**
*The robot always arrives at the goal by using RobotPathGeneration algorithm as long as the robot is not completely surrounded by obstacles.*


**Proof Theorem** **4.**In the first case (see Figure 7), the lower water level cell is certainly being presented to the four cells adjacent to the robot’s position. Thus, there is a water level difference, and so the robot should move to the lower water level cell. This is contradictory and so the assumption is false. In the case of a circular path (see Figure 8), we can arbitrarily select a cell and set it to be the start cell for the robot’s motion loop, and its movement along the circular path causes the water level to become lower. However, on some occasions, the robot might jump to the start cell, but it is restricted from moving to a higher water level cell in the water sink model. This is contradictory and so the assumption is false. □

In a world with obstacles, there is a path from the robot’s position to the plughole (goal) by Theorem 2 and the robot can reach the goal by Theorems 3 and 4.

## 4. Experimental Results

### 4.1. Implementation of Water Sink Model

#### 4.1.1. Experimental Simulator Setup

An experimental simulator was implemented with Visual C++ (Visual Studio Community 2015) on Windows OS (see Figure 9). The simulator is composed of three parts. The first is map generation part (see Figure 10a). At this part, operator can create new map with any size and load or save a map. The second part is a potential field part (see Figure 10b). This part is needed to compare with the water sink model and the operator determines the potential field parameters. The last part is the water sink model part (see Figure 10c). The operator can set turn number for drainage, but this is not the main function of this part and the water sink model has one input parameter, which is the amount of water of drainage per one turn.

#### 4.1.2. The Process of Water Sink Model Simulation

The implemented simulator can make a water map with the cell size and the number of cells. After that, the user can determine obstacle (black-colored cell), robot position (pink-colored cell) and plug hole (white-colored cell). When the cell in which the robot is located is involved in the drainage, the draining is completed, and the robot starts to move. When the robot reaches the plughole, the experiment is terminated. The robot steps by water sink are colored dark-pink cell (see Figure 11).

In the process of simulation, the robot is assumed as a point and to be always positioned at the center of a cell. A cell is considered to be an obstacle cell if the cell contains even a part of obstacle. Since the water sink model generates a path on the grid metric, there are some stairway paths or sharp turns. Even though the robot is assumed as a point in this implemented simulator, these stairway paths or sharp turns may present another problem deriving from the kinodynamic characteristics of the robot. In this case, some additional path smoothing techniques can resolve it [21,22].

#### 4.1.3. Several Other Results

To confirm the accuracy and practicality of the water sink model, some sample maps are generated with 10 (pixels) size and 60 × 60 cells. The drainage water is 1000 per turn. The experimental results of the water sink model are shown in Figure 12.

The several other results of the potential field are shown in Figure 13. In this experiment, the source maps are the same as the water sink model. However, there are no results of reaching to the goal in potential field method. The parameter ξ has variation from 0.01 to 0.02, η has variation from 1.0 to 2.5, and $\rho_{0}$ has variation from 10.0 to 155.5.

The potential field method has three parameters; ξ, η and $\rho_{0}$. Figure 14 shows the results of the variation of parameters. The grid map is the same as Figure 12a and Figure 13a. Here, the parameter value that is considered to be sufficiently small and large is respectively set. The parameters should be empirically found with the combination of each parameter characteristic (see Table 1).

#### 4.1.4. Water Sink Model for Dynamic Environment

For the application of the water sink model in a dynamic environment, two additional assumptions are required. The first assumption is that the robot knows the goal position. The second assumption is that the robot can store the obstacle location whenever the robot encounters or find out some obstacles. It is also assumed that the entire outside of the sensing area of the robot is already filled with water. The robot could move using the water sink model after drainage. The drainage will occur again and the obstacle location will be memorized in the robot when the robot encountered some obstacles after movement. Finally, the robot could repeatedly reach the goal with the above process. Figure 15 shows a simple example of simulation for the application to dynamic environment. There is a 7 × 4 sized obstacle and the sensing area of the robot is assumed as 7 × 7 cells around the robot (see Figure 15a). The water drainage occurred once at the first time as the determination of the robot path. However, the sensor of robot could not detect any obstacle the first time (see Figure 15b). After a few movements of robot, the robot finds out some obstacles (see Figure 15c). However, the obstacles do not yet interfere with the robot path. Therefore, the robot continues to move. In Figure 15d, the robot encounters the obstacle that disturbs the determined robot path. At this time, the water drainage occurred again based on the current information of the robot with obstacles. In Figure 15e, the obstacle disturbs the robot path and the water drainage occurred again at this time. Finally, the robot modifies the path and it could reach the plughole (see Figure 15f).

Figure 16 shows another example for the application to the dynamic environment. The map is more complex than Figure 15 and the same as Figure 12a. The sensing area of the robot is assumed as 11 × 11 cells around the robot (see Figure 16a). The water drainage occurred once at the first time (see Figure 16b) and the robot moves to the plughole without recognition of any obstacle at first. However, whenever the robot encounters an obstacle at the next robot position, local water drainage occurred again, from the plughole, not to the whole area, but to the current robot position (see Figure 16c–f). Figure 16g,h shows that the robot could finally reach the plughole in the dynamic environment. Here, the water drainage totally occurred five times and the path length is 62.

### 4.2. Comparision with Potential Field

#### 4.2.1. Experimental Setup for Potential Field Method

For the comparison of the water sink model and the potential field method, the test maps, including various shapes and numbers of obstacles, are needed. However, the local minima exist in almost cases of potential field map. Therefore, some simple maps (see Figure 17) with various parameters set were used. Table 2 shows the parameter set and 12 combinations of parameters are used in each map.

#### 4.2.2. The Results of Water Sink Model

The water sink model has no parameter and the path is deterministic at any case. Figure 18 shows the water sink model results at each map. The robot path length is 23 on map-1 and 57 on map-2. The path lengths of sideways, upside down, and diagonal are all equal to 1.

#### 4.2.3. The Results of Potential Field Method and Comparison

The goal(plughole) reaching of a potential field method is dependent on the parameters (see Figure 19). The simulation is performed on a parameter set. Table 3 and Table 4 show the success or failure and the path length of robot on the map-1 and map-2. Table 5 shows the average path length of the potential field when the robot reached the goal (plughole) successfully, the path length of water sink model and the path length ratio.

### 4.3. Comparision with Potential Field Using Random Walk

It is easily found that one or more local minima always occur in the potential field method if the complexity of the map is more than a certain level. To overcome this problem, the random walk method could be applied in addition to the potential field method. The random walk method is one of the probability-based techniques. Although the random walk technique does not guarantee processing time and path length of the robot, but the robot reached the goal in most times of our experiments. When the robot encounters some local minima, the robot randomly determines a direction and it moves to escape the local minima as pre-defined length. In our experiment, the pre-defined length is assumed as five cells and the direction is randomly determined without consideration of obstacles and goal location.

Figure 20 shows the results of potential field with random walk as various parameter changes. Each experiment was performed 10 times and the path length is selected as the shortest one. Here, the purple colored cell means the trace of random walk. Unfortunately, it is very hard to find some tendency of parameter change effects, as in Figure 20. For example, the path length was increased in Figure 20c when compared with Figure 20a, as ξ increases when all other parameters are the same, but the path length was decreased in Figure 20d as compared with Figure 20b as ξ increases. Similarly, the effects of parameter η and $\rho_{0}$ also show no tendency. In addition, the variation of path length as the parameter changes was high.

For the comparison with potential field using the random walk and water sink model, some experiments were performed in the same maps that were used in Figure 14 and Figure 15. The parameters of the potential field method with random walk were ξ=0.05, η=2.1, $\rho_{0}=65.5$ for Figure 21a,b,d,e, ξ=0.1, η=5.0, $\rho_{0}=100.0$ for Figure 21g,h and ξ=0.05, η=2.1, $\rho_{0}=55.0$ for Figure 21j,k. These parameters were empirically selected by repetitive experiments that consider some parameter range. The used ranges of parameters were [0.01, 0.1] for ξ, [1.0, 5.0] for η and [40.0, 150.0] for $\rho_{0}$. The results of the first column in Figure 21 are the case of shortest path length and the results of the second column in Figure 21 are the case of the longest path length in our experiments on each map. The results of the last column in Figure 21 are the result of the water sink model method on each map. Each path length is described in the caption of Figure 21.

In the results of the potential field method, it could be easily found that the path length could be too much large, even though the complexity of map is simple. In addition, there is a tendency to renew the maximum path length whenever more experimental results are updated. On the other hand, the path length of water sink model is always smaller than the path length of the potential field method and its value remains constant.

## 5. Conclusions

In this paper, we propose a water sink model to resolve the local minima problem found in previous potential field methods for path planning. The water in the field drains from a higher location to a lower one that is affected by gravity. Basically, the proposed method mimics the draining of water from a sink with a plughole. As a result, when compared to the potential field, the proposed method achieved a shorter path with low clearance from obstacles and without local minima. In addition, it is beneficial for robot path planning, since it does not require an additional controller similar to potential field methods.

The water sink model is provided to resolve the local minima problem of potential field method, but there are some considerations in the application of the water sink model to real world problems. The first one is that the robot is assumed as a point in water sink model. Commonly, the robot has own some mechanical and kinodynamic characteristics. Therefore, the post-processing for path smoothing is additionally required when considering the kinodynamic characteristics of the robot after path planning. The second one is that the obstacle is approximated as a rectangle with sharp edge, because the water sink model is based on the grid cell. In the water sink model, the obstacle region is recognized and processed, as it is greater than or equal to the actual obstacles. Since the size of the cell may not be sufficiently small, the path may be clogged due to an over-approximation of the obstacle, even though there is actually empty space for the robot to pass through. On the other hand, if the size of the cell is small enough, the resolution of representation for the problem space may be improved, but the time for processing will be increased enormously. For the same reason that the obstacle is approximated as a grid cell, the robot is moving along the obstacles when the robot encounters the obstacle. Because of this, the clearance issue should be considered. The third one is that, since the water sink model is based on grid cell space, grid cells also present the obstacles and the sequence of path. To more accurately improve the representation of the real world, the hexagonal cell or triangular mesh could be considered for the representation of the real problem space. Especially, applying the irregular triangle mesh that can more accurately represent the real world and reduce the processing time like the non-uniform cell decomposition method.

Nonetheless, the water sink model is very simple and intuitive in finding the path. Additionally, the water sink model is a kind of global path planning method and it could always find the shortest path in the workspace.

## Figures and Tables

**Figure 1 sensors-19-01269-f001:**
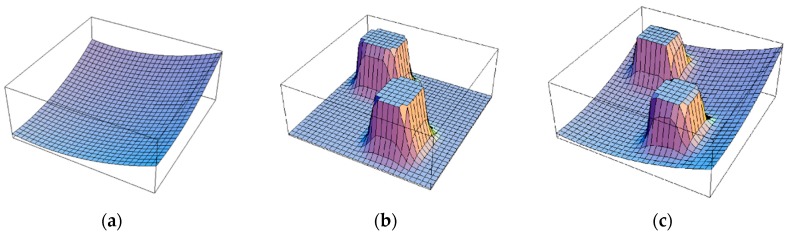
The potential field method: (**a**) attractive forces; (**b**) repulsive forces; and, (**c**) the sum of attractive forces and repulsive forces for potential filed.

**Figure 2 sensors-19-01269-f002:**
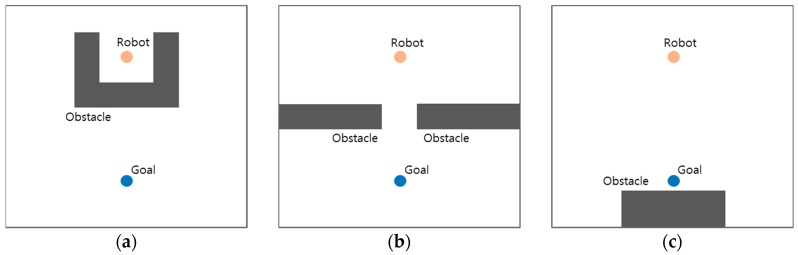
Representative local minima in a potential field: (**a**) the robot is surrounded by obstacles and the exit is opposite the goal; (**b**) narrow distance between obstacles; and, (**c**) the goal is very close to the obstacles.

**Figure 3 sensors-19-01269-f003:**
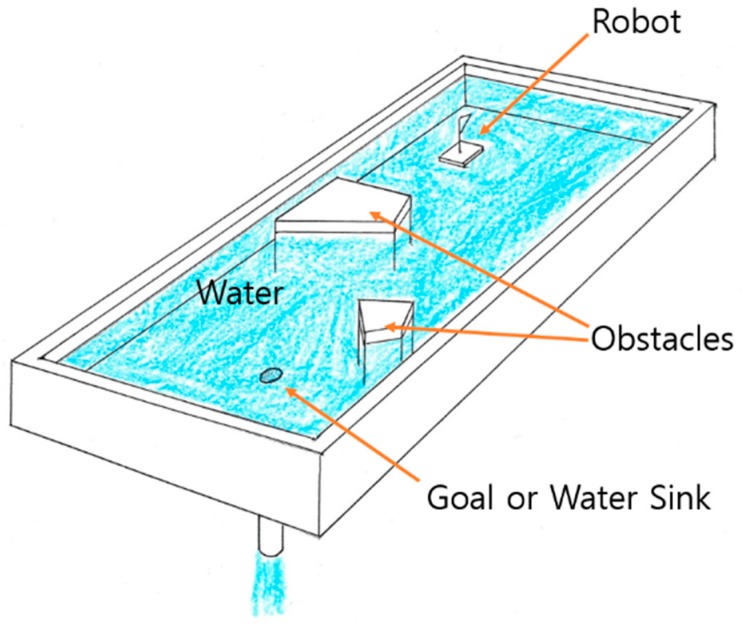
Water sink model concept.

**Figure 4 sensors-19-01269-f004:**
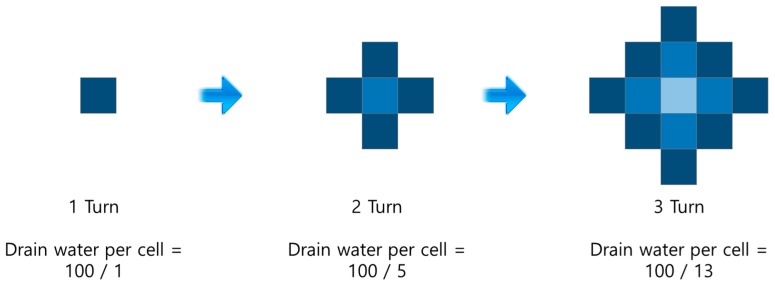
The water sink grid model.

**Figure 5 sensors-19-01269-f005:**
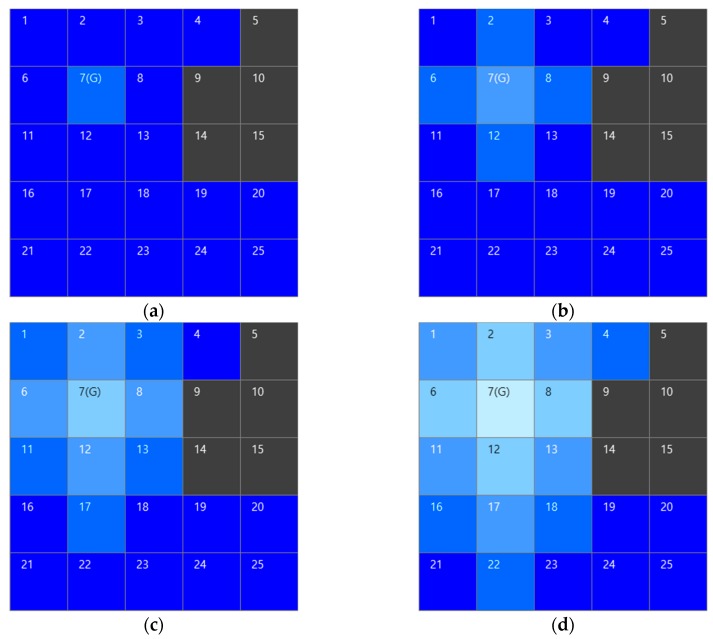
The sequences of water flow in water sink model. The cell number 7(G) is goal or plug hole and cell number 5, 9, 10, 14, and 15 are obstacles. These examples show the progress of water drainage as: (**a**) first turn, only plug hole cell is concerned about drainage; (**b**) second turn, plug hole, and the adjacent cells are involved in drainage; (**c**) third turn; and, (**d**) fourth turn, the cells that are involved in the drainage gradually expand and the each cell color becomes gradually brighter.

**Figure 6 sensors-19-01269-f006:**
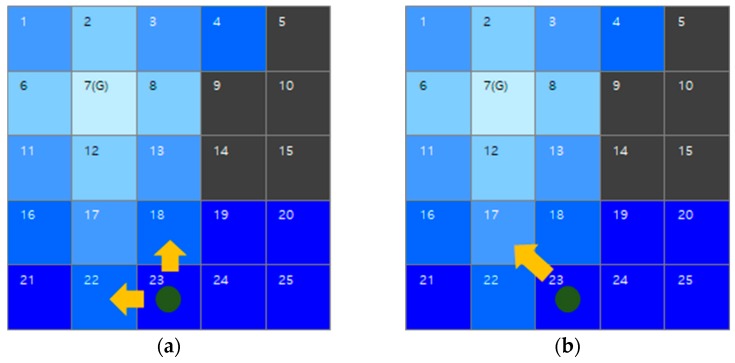
An example of water sink model-based path generation when using four-connectivity and eight-connectivity: (**a**) the case of four-connectivity for robot movement; (**b**) the case of eight-connectivity for robot movement.

**Figure 7 sensors-19-01269-f007:**
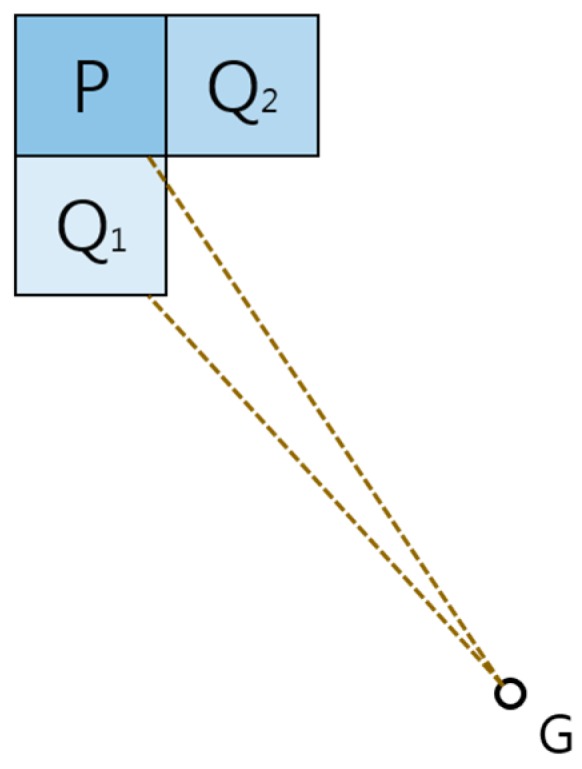
The case that robot is stopped in another position P, not same with position G.

**Figure 8 sensors-19-01269-f008:**
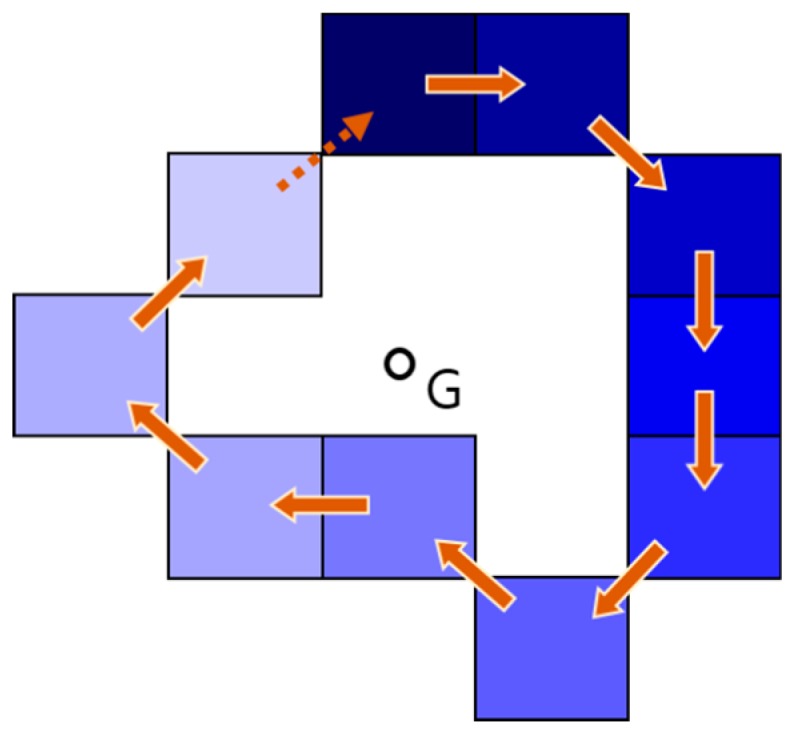
The case of circular path.

**Figure 9 sensors-19-01269-f009:**
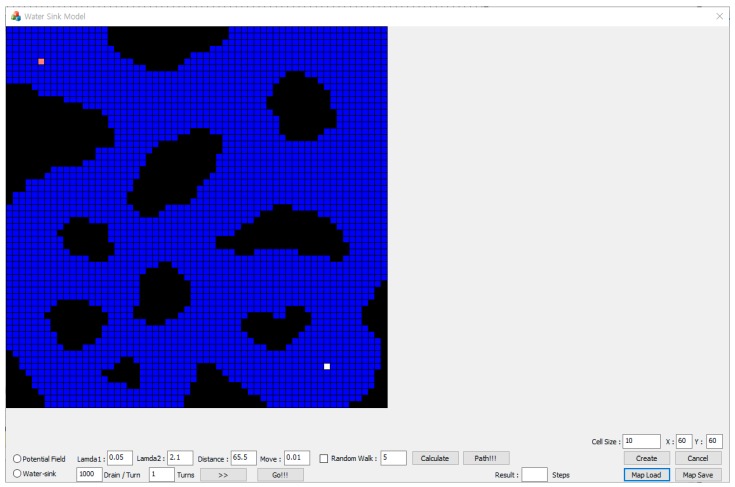
Water sink model simulator.

**Figure 10 sensors-19-01269-f010:**
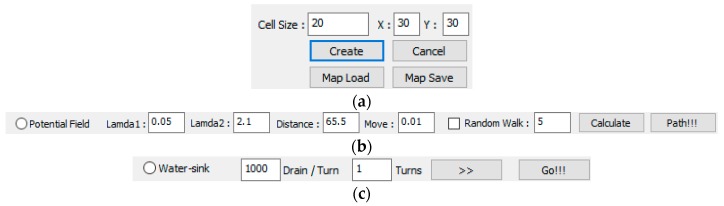
Parameter adjustment of experimental simulator: (**a**) map creation part; (**b**) potential field part; and, (**c**) water sink model part.

**Figure 11 sensors-19-01269-f011:**
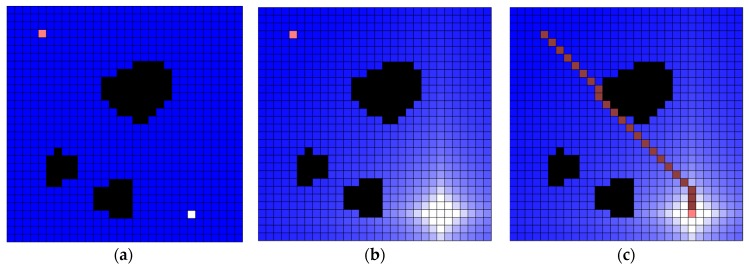
Water sink model simulator example. Example is 20 (pixels) cell size, 30 × 30 cells and the following is a series of map creation, water drainage, and robot path. The drainage water is 1000 per turn: (**a**) there are three obstacles, the robot is located at the upper-left corner, and the plughole is located at the lower-right corner; (**b**) water drainage progressed up to 50 turns; and, (**c**) final result of robot path.

**Figure 12 sensors-19-01269-f012:**
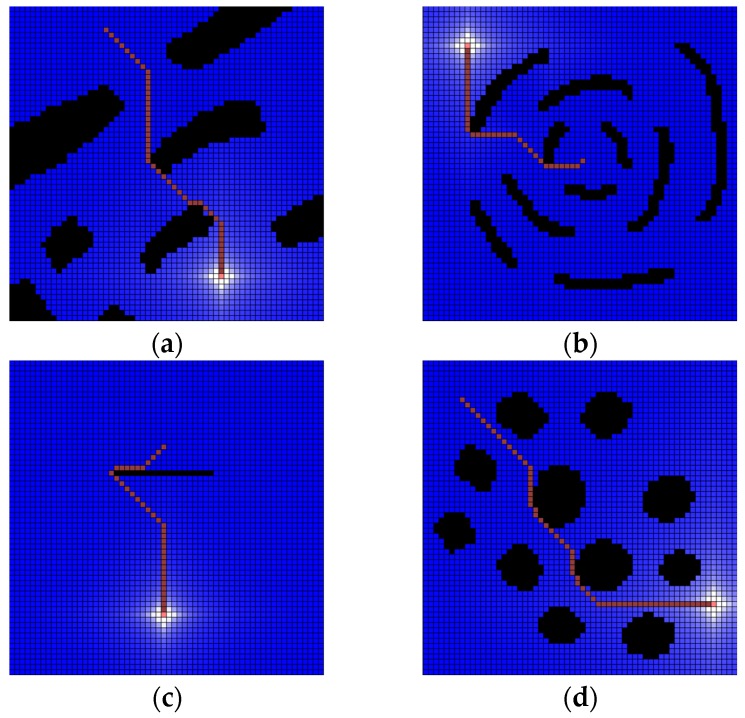
Several results of water sink model. The cell size is 10 (pixels) and 60 × 60 cells map. The drainage water is 1000 per turn. All experiments show that the robot (pink-colored cell) reached the plughole (white-colored cell) with the shortest path on the grid metric. As a result, the plughole is obscured by the robot: (**a**) test map-1 for water sink model; (**b**) test map-2 for water sink model; (**c**) test map-3 for water sink model; (**d**) test map-4 for water sink model.

**Figure 13 sensors-19-01269-f013:**
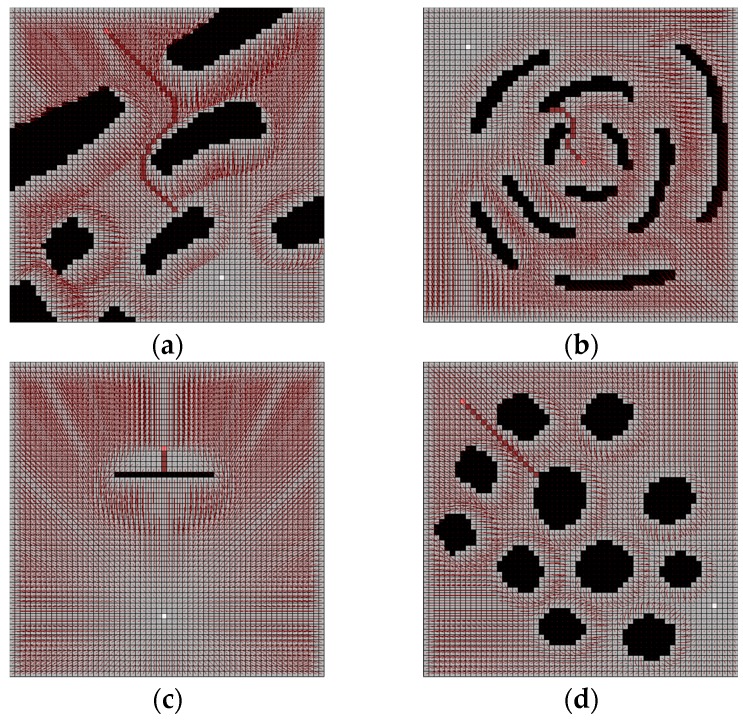
Several results of potential field method with the same map of water sink model in Figure 12: (**a**) test map-1 for potential field method; (**b**) test map-2 for potential field method; (**c**) test map-3 for potential field method; (**d**) test map-4 for potential field method.

**Figure 14 sensors-19-01269-f014:**
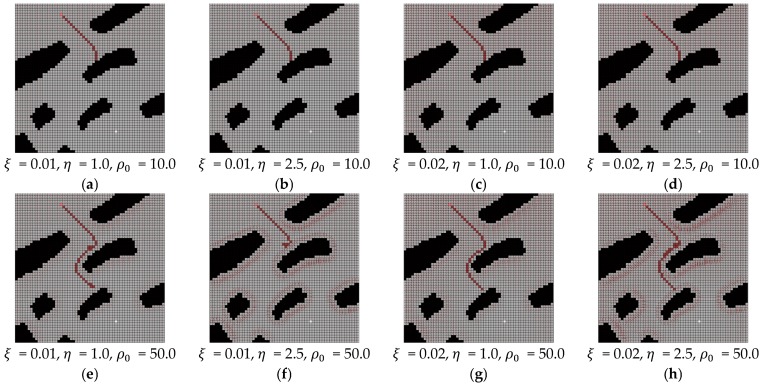
The results of potential field method with parameter variations.

**Figure 15 sensors-19-01269-f015:**
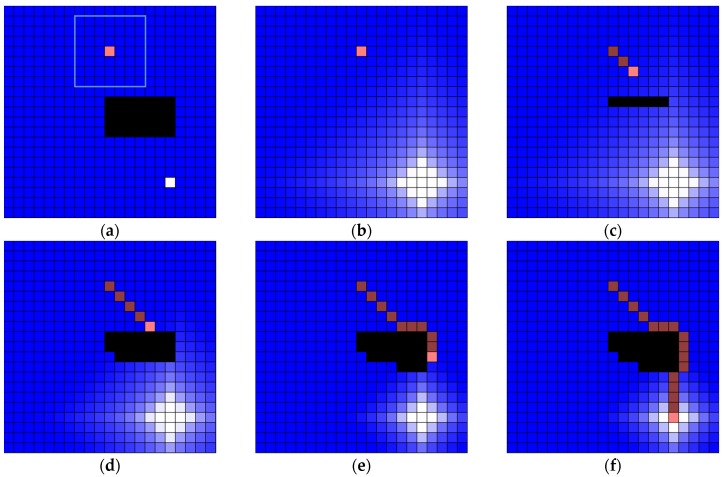
An example of simulation for dynamic environment. Workspace includes 21 × 21 cells whose size is 20 pixels: (**a**) there is a 7 × 4 sized obstacle and the sensing area of robot is 7 × 7 cells; (**b**) the sensor of robot could not detect any obstacle and water drainage is occurred at first; (**c**) the obstacle could not interfere with the robot path; (**d**) the robot path is disturbed by an obstacle and water drainage is occurred again; (**e**) the robot path is disturbed by obstacle again and water drainage is occurred again; and, (**f**) the final result of water sink model.

**Figure 16 sensors-19-01269-f016:**
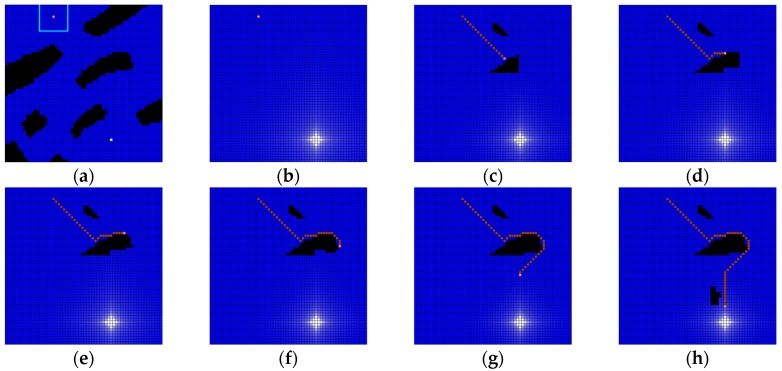
Another example of simulation for dynamic environment. Workspace includes 60 × 60 cells whose size is 10 pixels: (**a**) there are some obstacles and the sensing area of robot is 11 × 11 cells; (**b**) the sensor of robot could not detect any obstacle and water drainage is occurred at first; (**c**) the robot encounters some obstacles at first; (**d**–**f**) the robot path is disturbed by an obstacle and water drainage is occurred again; (**g**) the robot escapes the obstacle area; and, (**h**) some obstacles are found, but there is no interference with the robot path. Finally, the robot could reach the plughole.

**Figure 17 sensors-19-01269-f017:**
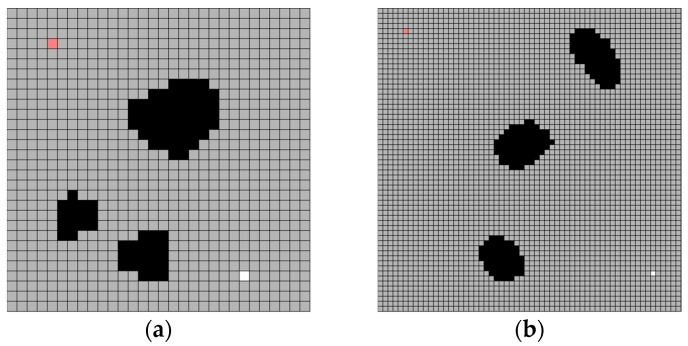
The potential field maps for comparison with water sink model: (**a**) map-1: 30 × 30 size map; and, (**b**) map-2: 60 × 60 size map.

**Figure 18 sensors-19-01269-f018:**
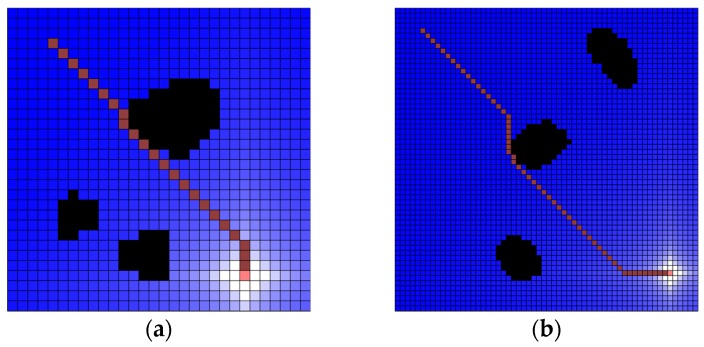
The results of water sink model. The path length is (**a**) map-1: 23, the same as Figure 10c; and, (**b**) map-2: 57.

**Figure 19 sensors-19-01269-f019:**
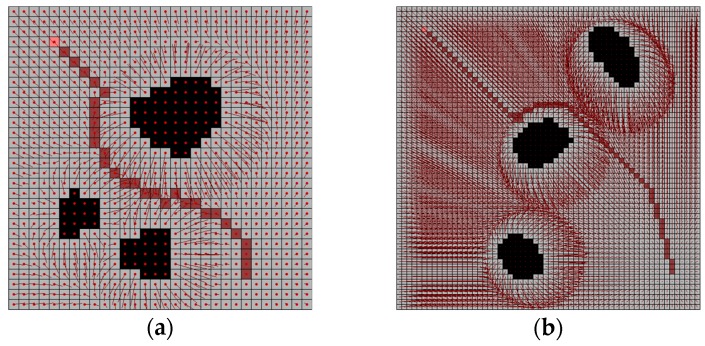
The results of water sink model. The path length is: (**a**) map-1: 23; and, (**b**) map-2: 57.

**Figure 20 sensors-19-01269-f020:**
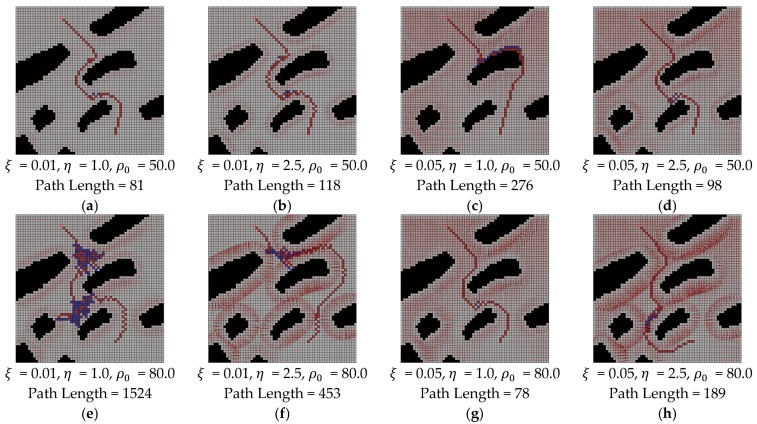
The results of potential field method using random walk with parameter variations.

**Figure 21 sensors-19-01269-f021:**
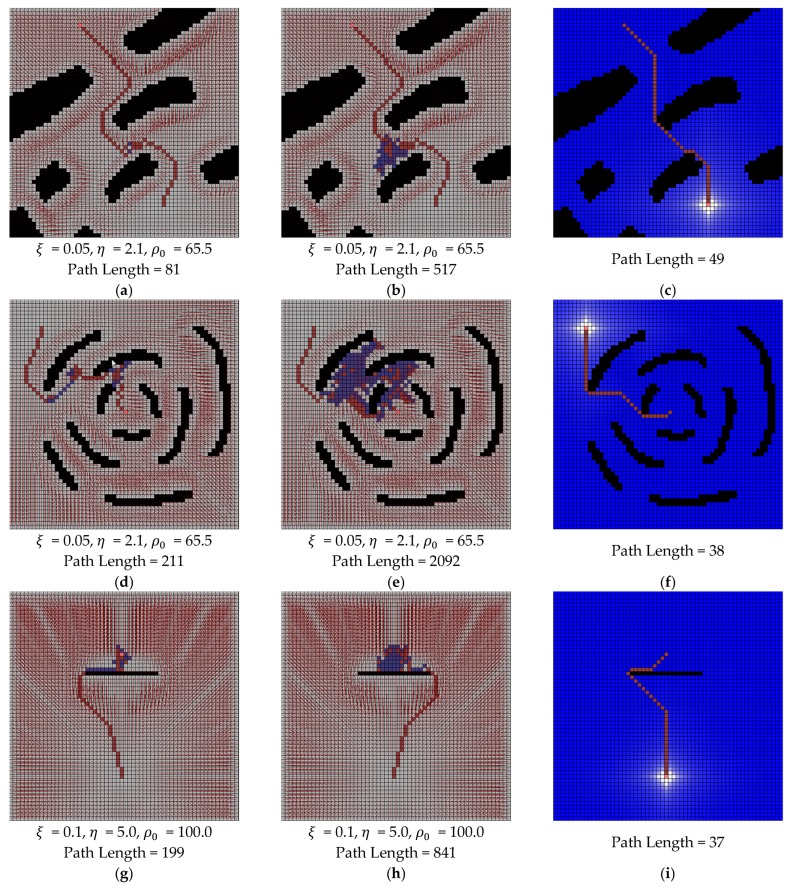
The results of comparison between water sink model and potential field with random walk: (**a**,**b**,**d**,**e**) parameters are ξ=0.05, η=2.1 and $\rho_{0}=65.5$; (**g**,**h**) parameters are ξ=0.1, η=5.0 and $\rho_{0}=100.0$; (**j**,**k**) parameters are ξ=0.05, η=2.1 and $\rho_{0}=55.0$; (**c**,**f**,**i**,**l**) are the results of water sink model.

**Table 1 sensors-19-01269-t001:** The characteristics of parameters for potential field method.

Parameters	ξ	η	$\rho_{0}$
small value	The robot could not move from the beginning	The robot could move around obstacles	Fail to reflect overall obstacle situation
large value	The robot could stop when encountering an obstacle	The robot could move to opposite to the goal	The robot could be affected by obstacles in distant

**Table 2 sensors-19-01269-t002:** Parameter set for map-1 and map-2.

Map	ξ	η	$\rho_{0}$
map-1	0.05, 0.1	2, 5	50, 100, 150
map-2	0.05, 0.1	2, 3	50, 70, 90

**Table 3 sensors-19-01269-t003:** Path length on map-1 for potential field method.

ξ	η	$\rho_{0}$	Result/Path Length
0.05	2	50	NR ^1^
0.05	2	100	23
0.05	2	150	26
0.05	5	50	23
0.05	5	100	25
0.05	5	150	29
0.1	2	50	NR
0.1	2	100	NR
0.1	2	150	23
0.1	5	50	NR
0.1	5	100	23
0.1	5	150	27

^1^ Not Reached.

**Table 4 sensors-19-01269-t004:** Path length on map-2 for potential field method.

ξ	η	$\rho_{0}$	Result/Path Length
0.05	2	50	NR ^1^
0.05	2	70	NR
0.05	2	90	NR
0.05	3	50	59
0.05	3	70	NR
0.05	3	90	NR
0.1	2	50	NR
0.1	2	70	NR
0.1	2	90	60
0.1	3	50	NR
0.1	3	70	59
0.1	3	90	63

^1^ Not Reached.

**Table 5 sensors-19-01269-t005:** Comparison of water sink model with potential field method. Averaged path length of potential field, if successful, deterministic path length of water sink model and path length ratio. The unit is cell.

Map	Average Path Length of Potential Field (PF)	Path Length of Water Sink Model (WSM)	Path Length Ratio (WSM/PF)
map-1	24.875	23	0.9246
map-2	60.25	57	0.9461

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
