# Peer review of "Water Sink Model for Robot Motion Planning"

_sensors, 2019, doi:10.3390/s19061269_

Round 1
Reviewer 1 Report
The authors’ purpose is to present a robot motion planning method, namely water sink model-based motion planning, to resolve the local minima problem using the potential field method by mimicking the flow of water draining out of a sink through a plughole. In their model, when the plug is removed, water starts to drain out via the plughole and the robot can reach the goal by the water flow.
In my opinion, this is a weak work. In general, the paper is badly presented, and the contribution of the paper is not clear; the idea does not seem to be novel. The paper lacks experiments, analysis and comparative. The authors need to show significant improvement over other approaches to claim the advancement and novelty of their proposal.
There are some points that must be taken to improve the quality of the article:
* The related work section must be focus on related work (state of the art) following the topic presented in this article (robot motion planning). It should include advantages, disadvantages, issues, etc.
* Make a well-suited citation. For a research paper, the following forms of the citation must be avoided:
“… to the goal [7-10]. All of these …”, “… local minima problem [11-17] caused by …”.
The cited related work should include advantages, disadvantages, issues, etc.
* Figures 5 and 14 must be referenced in the text. These figures must be mentioned and discussed in the text.
* In the proposal (Algorithm 1 and 2). How the mobile robot configuration and dimensions are considered? Also, how the environment configuration (position and size of the obstacles) is considered?
* The proposal presents a robot motion planning method for static environments, but state of the art motion planning methods present solutions for static and dynamic environments. How your proposal can deal with dynamic environments. The proposal must consider dynamic environments to provide advancement and novelty.
* In section 4, how were established the drainage water per turn value of 1000, and the parameter’s range gamma, eta and rho zero. How would the results of your proposal vary with changing these values?
* In general, there is a lack of parameter setting discussion. How were chosen the parameters and its values? How would the results of your proposal vary with increasing/decreasing these values?
* In the results, there is not a comparison with respect to the state of the art. The results are insufficient to provide a valid evaluation of the proposed method.
* The experiments are insufficient. The article presents a weak comparison between the water sink model and the potential field method. However, it should be performed a deep comparative study with several scenarios and state of the art motion planning methods.
* The "Conclusions" section should be improved to provide real useful conclusions. Not to be an overview of the previous paragraphs.
* As a recommendation, on the conclusions, the authors need to clearly provide solid and insightful future research directions in a separate paragraph.
* References must be updated, and its number should be increased. A review of the state of the art is required.
Author Response
Response to the reviewer 1
The authors’ purpose is to present a robot motion planning method, namely water sink model-based motion planning, to resolve the local minima problem using the potential field method by mimicking the flow of water draining out of a sink through a plughole. In their model, when the plug is removed, water starts to drain out via the plughole and the robot can reach the goal by the water flow.
In my opinion, this is a weak work. In general, the paper is badly presented, and the contribution of the paper is not clear; the idea does not seem to be novel. The paper lacks experiments, analysis and comparative. The authors need to show significant improvement over other approaches to claim the advancement and novelty of their proposal.
There are some points that must be taken to improve the quality of the article:
The original comments used ‘*’ as an identifier, the number have been added for convenience.
1. The related work section must be focus on related work (state of the art) following the topic presented in this article (robot motion planning). It should include advantages, disadvantages, issues, etc.
As following the review comments, the contents are added and modified, Line : 36-42, 46-62, 252-256
2. Make a well-suited citation. For a research paper, the following forms of the citation must be avoided:
“… to the goal [7-10]. All of these …”, “… local minima problem [11-17] caused by …”.
The cited related work should include advantages, disadvantages, issues, etc.
As following the review comments, the contents are corrected and added, Line : 36-42, 46-62
3. Figures 5 and 14 must be referenced in the text. These figures must be mentioned and discussed in the text.
As following the review comments, the references are added, Line : 119-121
4. In the proposal (Algorithm 1 and 2). How the mobile robot configuration and dimensions are considered? Also, how the environment configuration (position and size of the obstacles) is considered?
As following the review comments, the contents are added, Line : 251-252
5. The proposal presents a robot motion planning method for static environments, but state of the art motion planning methods present solutions for static and dynamic environments. How your proposal can deal with dynamic environments. The proposal must consider dynamic environments to provide advancement and novelty.
As following the review comments, the contents are added, Line : 257-263
6. In section 4, how were established the drainage water per turn value of 1000, and the parameter’s range gamma, eta and rho zero. How would the results of your proposal vary with changing these values?
Regardless of drain value, the water sink model works. In our implementation, the water value is used for displaying the cell brightness. The displayable cell color range is 0-255. Actually, the drainage water per turn is added to the color value of cell. In case of the drain value is 1000, the plug-hole color is saturated at the first step. When the drain value is large, the color of cells is white rapidly. Contrary the drain value is small, the color of cells hardly changes.
The parameters of potential field method are tuned by empirically.
As following the review comments, the results of changing of the parameters of potential field method are added, Line : 276-280
7. In general, there is a lack of parameter setting discussion. How were chosen the parameters and its values? How would the results of your proposal vary with increasing/decreasing these values?
As following the review comments, the contents are added, Line : 281-282
8. In the results, there is not a comparison with respect to the state of the art. The results are insufficient to provide a valid evaluation of the proposed method.
As following the review comments, the references are added and modified, Line : 49-56
9. The experiments are insufficient. The article presents a weak comparison between the water sink model and the potential field method. However, it should be performed a deep comparative study with several scenarios and state of the art motion planning methods.
As following the review comments, the references are added and modified, Line : 49-56
10. The "Conclusions" section should be improved to provide real useful conclusions. Not to be an overview of the previous paragraphs.
As following the review comments, the references are added and modified, Line : 313-327
11. As a recommendation, on the conclusions, the authors need to clearly provide solid and insightful future research directions in a separate paragraph.
As following the review comments, the references are added and modified, Line : 328-340
12. References must be updated, and its number should be increased. A review of the state of the art is required.
As following the review comments, the references are added and modified, Line : 36-42, 46-62, 252-256, 373-377, 382-386, 389-393, 398-403
Reviewer 2 Report
Comments:
Authors have proposed a method to overcome the problem of getting stuck in local minima in potential field planners. It uses a water sink model to mimic the water flow with a plughole and a floating piece. Simulation results are provided and comparison is done with potential field.
Overall, the idea is interesting. However, there appears to be flaws in the method. The flaws mainly comes from very simple assumptions.
1.
Theorem 1.
"The water level of a cell further from the plughole is always higher than one closer to the plughole".
How is this a theorem? This is an assumption in author's work. There is a concrete meaning and differences between a lemma, theorem, proposition, corollary, and assumption. What is presented here simple draws from an assumption.
2.
Moreover, the proof of Theorem 1 does not make any sense.
Eqs.4 and 5 are not at all related to Eq.6, 7, ....
3.
What is 'C' and 'a' in Eq.6 and 7?
4.
Lines 131 - 133: (0 ≤ d ≤ t) .... (0 ≤ d1 < d2 ≤ t)
'd' is distance, and 't' is time. So, d <= t, makes no sense. Their units are different. It is like saying, 2 meters < 5 sec.
5.
Eq.8, Vd1(t) < Vd2(t). This is an assumption in author's work. Nothing to prove here.
6.
Eq.8, Vd1(t) < Vd2(t) for d1 < d2. And d is the distance from the plughole.
Do the authors mean to say that as distance from the plughole increases, the height also increases? In other words, the plughole is situated at a lower height that other areas. Farthest areas are at the most height?
This is a very simplistic assumption. In real world, the configuration of obstacles etc. are too random. The local minima can occur as small 'mud-pockets' in authors analogy.
7.
Theorem 2: there is no local minima.
Again this is not a theorem but an assumption that going far from the sinkhole, the height always increases, so there are no flat areas. Therefore, Theorem 3 and 4 also follows and are the same.
8.
Looking at the results, the search space of the algorithm is huge. It is similar to Dijkstra's algorithm and basically searches the entire map. Ex. see Fig.11(c) for a simple obstacle case. This is not optimal at all. This seems to be like, make a potential path, also make a Dijkstra path. If you are stuck on the potential path, them keep moving along the Dijkstra path.
9.
It is not clear if the paths generated by the algorithm are smooth. The results show sharp turns at some places. Authors should include a point about the smoothness of the path. Authors should also include references of smoothing techniques with some latest review works in path smoothness and safety of robots like:
[1]
"A Review of Motion Planning Techniques for Automated Vehicles" David Gonzalez Bautista et. al.
IEEE Transactions on Intelligent Transportation Systems · November 2015 DOI: 10.1109/TITS.2015.2498841
[2]
Ravankar, A.; Ravankar, A.A.; Kobayashi, Y.; Hoshino, Y.; Peng, C.-C.
"Path Smoothing Techniques in Robot Navigation: State-of-the-Art, Current and Future Challenges."
Sensors 2018, 18, 3170. DOI: 10.3390/s18093170.
10.
In addition, the entire introduction section needs to be improved. More recent related papers needs to be included in the paper and previous works should be discussed.
11.
While the idea is interesting, I think that the authors need to do more experiments in complex environments, give the comparison in terms of processing time, evaluate the complexity of the algorithm, and revise accordingly.
Author Response
Response to the reviewer 2
Authors have proposed a method to overcome the problem of getting stuck in local minima in potential field planners. It uses a water sink model to mimic the water flow with a plughole and a floating piece. Simulation results are provided and comparison is done with potential field.
Overall, the idea is interesting. However, there appears to be flaws in the method. The flaws mainly comes from very simple assumptions.
1.
Theorem 1.
"The water level of a cell further from the plughole is always higher than one closer to the plughole".
How is this a theorem? This is an assumption in author's work. There is a concrete meaning and differences between a lemma, theorem, proposition, corollary, and assumption. What is presented here simple draws from an assumption.
As following the review comments, the contents are added and modified, Line : 166-175, 177-184
2.
Moreover, the proof of Theorem 1 does not make any sense.
Eqs.4 and 5 are not at all related to Eq.6, 7, ....
As following the review comments, the contents are modified, Line : 177-184
3.
What is 'C' and 'a' in Eq.6 and 7?
As following the review comments, the contents are modified, Line : 177-184
4.
Lines 131 - 133: (0 ≤ d ≤ t) .... (0 ≤ d1 < d2 ≤ t)
'd' is distance, and 't' is time. So, d <= t, makes no sense. Their units are different. It is like saying, 2 meters < 5 sec.
As following the review comments, the contents are added and modified, Line : 122-123, 177-184
5.
Eq.8, Vd1(t) < Vd2(t). This is an assumption in author's work. Nothing to prove here.
As following the review comments, the contents are modified, Line : 177-184
6.
Eq.8, Vd1(t) < Vd2(t) for d1 < d2. And d is the distance from the plughole.
Do the authors mean to say that as distance from the plughole increases, the height also increases? In other words, the plughole is situated at a lower height that other areas. Farthest areas are at the most height?
This is a very simplistic assumption. In real world, the configuration of obstacles etc. are too random. The local minima can occur as small 'mud-pockets' in authors analogy.
As following the review comments, the contents are modified, Line : 177-184
7.
Theorem 2: there is no local minima.
Again this is not a theorem but an assumption that going far from the sinkhole, the height always increases, so there are no flat areas. Therefore, Theorem 3 and 4 also follows and are the same.
As following the review comments, the contents are modified, Line : 186-187, 201-202, 209-210
8.
Looking at the results, the search space of the algorithm is huge. It is similar to Dijkstra's algorithm and basically searches the entire map. Ex. see Fig.11(c) for a simple obstacle case. This is not optimal at all. This seems to be like, make a potential path, also make a Dijkstra path. If you are stuck on the potential path, them keep moving along the Dijkstra path.
As following the review comments, the contents are added, Line : 119-123, 143-149
9.
It is not clear if the paths generated by the algorithm are smooth. The results show sharp turns at some places. Authors should include a point about the smoothness of the path. Authors should also include references of smoothing techniques with some latest review works in path smoothness and safety of robots like:
[1]
"A Review of Motion Planning Techniques for Automated Vehicles" David Gonzalez Bautista et. al.
IEEE Transactions on Intelligent Transportation Systems · November 2015 DOI: 10.1109/TITS.2015.2498841
[2]
Ravankar, A.; Ravankar, A.A.; Kobayashi, Y.; Hoshino, Y.; Peng, C.-C.
"Path Smoothing Techniques in Robot Navigation: State-of-the-Art, Current and Future Challenges."
Sensors 2018, 18, 3170. DOI: 10.3390/s18093170.
As following the review comments, the contents are added, Line : 252-256
10.
In addition, the entire introduction section needs to be improved. More recent related papers needs to be included in the paper and previous works should be discussed.
As following the review comments, the contents are added and modified, Line : 36-42, 46-62, 255-259, 373-377, 382-386, 389-393, 398-403
11.
While the idea is interesting, I think that the authors need to do more experiments in complex environments, give the comparison in terms of processing time, evaluate the complexity of the algorithm, and revise accordingly.
As following the review comments, the contents are added, Line : 131-134, 158-162
Round 2
Reviewer 1 Report
In this reviewed version of the manuscript, the authors have addressed some of my suggestions and comments. However, in my opinion, there are still some comments and suggestions that must be taken into consideration to improve the quality of the article.
I must insist that there is a lack of experiments, analysis, and comparison. The authors need to show significant improvement over other approaches to claim the advancement and novelty of their proposal.
The proposal presents experiments for robot motion planning in static environments, but state of the art motion planning methods present solutions for static and dynamic environments. The paper needs to show experiments for robot motion planning in dynamic environments, i.e., test environments with dynamic obstacles and unknown and partially unknown environments for the robot.
The experiments are insufficient. The article presents a weak comparison between water sink model and potential field methods. The results must show a comparative with the state of the art motion planning methods. The results are insufficient to provide a valid evaluation of the proposed method. It should be performed a deep comparative study with several scenarios and state of the art motion planning methods.
Author Response
Thank you for your valuable comments. Attachment please find the response.

Reviewer 2 Report
Dear Editor,
Authors have satisfactorily answered the questions and revised the paper well. I think that it can be accepted for publication after proof-reading.
Best Regards,
Reviewer
Author Response
Thank you for your review.
Round 3
Reviewer 1 Report
The revised version has been improved in the aspects of experimentation. However, in my opinion, there are still some comments and suggestions that must be taken into consideration to improve the quality of the article.
There is still a lack of experiments, analysis, and comparison. The authors need to show significant improvement over other approaches to claim the advancement and novelty of their proposal.
The revised version presents just one experiment for a dynamic environment. The experimentation must be extended to environments with dynamic obstacles to expose the proposal to complex offline and online path planning problems. In general, the experimentation is insufficient to validate the proposal.
The article presents a weak comparison between water sink model and potential field methods. The results of the potential field method using random walk are insufficient. How did you choose the values of the scaling factor of attraction force, scaling factor of repulsive force, and the influence distance? Why are fixed these parameters? Any specific reason for choosing these values: 0.05, 2.1, and 65.5, respectively? How the performance of the potential field method using random walk vary with increasing/decreasing these values?
Author Response
Round 3 : Response to the reviewer 1
The revised manuscript is attached.
The number have been added for convenience.
The revised version has been improved in the aspects of experimentation. However, in my opinion, there are still some comments and suggestions that must be taken into consideration to improve the quality of the article.
There is still a lack of experiments, analysis, and comparison. The authors need to show significant improvement over other approaches to claim the advancement and novelty of their proposal.
1. The revised version presents just one experiment for a dynamic environment. The experimentation must be extended to environments with dynamic obstacles to expose the proposal to complex offline and online path planning problems. In general, the experimentation is insufficient to validate the proposal.
As following the review comments, the contents and the scenarios about dynamic environment of water sink model are added, Line : 307-321
2. The article presents a weak comparison between water sink model and potential field methods. The results of the potential field method using random walk are insufficient. How did you choose the values of the scaling factor of attraction force, scaling factor of repulsive force, and the influence distance? Why are fixed these parameters? Any specific reason for choosing these values: 0.05, 2.1, and 65.5, respectively? How the performance of the potential field method using random walk vary with increasing/decreasing these values?
As following the review comments, the contents are added, Line : 362-371 and 373-389